

# Associations between SNPs and vegetation indices: unraveling molecular insights for enhanced cultivation of tea plant (*Camellia sinensis* (L.) O. Kuntze)

Daria Kuzmina[1,2], Lyudmila S. Malyukova[1], Karina Manakhova[1,2], Tatyana Kovalenko[1,2], Jaroslava Fedorina[1,2], Aleksandra O. Matskiv[1], Alexey V. Ryndin[1], Maya V. Gvasaliya[1], Yuriy L. Orlov[3] and Lidiia S. Samarina[1,2]

[1] Federal Research Centre the Subtropical Scientific Centre of the Russian Academy of Sciences, Sochi, Russia
[2] Sirius University of Science and Technology, Sochi, Russia
[3] Institute of Biodesign and Complex Systems Modeling, Sechenov First Moscow State Medical University (Sechenov University), Moscow, Russia

Corresponding authors
Yuriy L. Orlov, orlov@bionet.nsc.ru
Lidiia S. Samarina, q11111w2006@yandex.ru

## ABSTRACT

**Background**. Breeding programs for nutrient-efficient tea plant varieties could be advanced by the combination of genotyping and phenotyping technologies. This study was aimed to search functional SNPs in key genes related to the nitrogen-assimilation in the collection of tea plant *Camellia sinensis* (L.) Kuntze. In addition, the objective of this study was to reveal efficient vegetation indices for phenotyping of nitrogen deficiency response in tea collection.

**Methods**. The study was conducted on the tea plant collection of *Camellia sinensis* (L.) Kuntze of Western Caucasus grown without nitrogen fertilizers. Phenotypic data was collected by measuring the spectral reflectance of leaves in the 350–1100 nm range calculated as vegetation indices by the portable hyperspectral spectrometer Ci710s. Single nucleotide polymorphisms were identified in 30 key genes related to nitrogen assimilation and tea quality. For this, pooled amplicon sequencing, SNPs annotation and effect prediction with SnpEFF tool were used. Further, a linear regression model was applied to reveal associations between the functional SNPs and the efficient vegetation indices.

**Results**. PCA and regression analysis revealed significant vegetation indices with high R2 values (more than 0.5) and the most reliable indices to select ND-tolerant genotypes were established: ZMI, CNDVI, RENDVI, VREI1, GM2, GM1, PRI, and Ctr2, VREI3, VREI2. The largest SNPs frequency was observed in several genes, namely *F3'5'Hb*, *UFGTa*, *UFGTb*, *4Cl*, and *AMT1.2*. SNPs in *NRT2.4*, *PIP*, *AlaDC*, *DFRa*, and *GS1.2* were inherent in ND-susceptible genotypes. Additionally, SNPs in *AlaAT1*, *MYB4*, and *WRKY57*, were led to alterations in protein structure and were observed in ND-susceptible tea genotypes. Associations were revealed between flavanol reflectance index (FRI) and SNPs in *ASNb* and *PIP*, that change the amino acids. In addition, two SNPs in *4Cl* were associated with water band index (WBI).

**Conclusions**. The results will be useful to identify tolerant and susceptible tea genotypes under nitrogen deficiency. Revealed missense SNPs and associations with vegetation indices improve our understanding of nitrogen effect on tea quality. The findings in our study would provide new insights into the genetic basis of tea quality variation

under the N-deficiency and facilitate the identification of elite genes to enhance tea quality.

# INTRODUCTION

Tea, derived from the perennial evergreen woody plant *Camellia sinensis* (L.) O. Kuntze, stands as one of the world's most consumed beverages, prized for its aromatic flavor and potential health benefits (*Samanta, 2020*; *Sánchez et al., 2020*). Tea has demonstrated numerous pharmacological properties, including antioxidant and anticancer effects, as well as the ability to reduce metabolic issues and prevent cardiovascular diseases (*Chan et al., 2011*; *Filippini et al., 2020*; *Brimson et al., 2023*). The secondary metabolites that determine the tea quality, such as theanine, caffeine, flavonoids, and amino acids, enhance the beneficial biological activities and taste of tea plants (*Gai et al., 2019*). The metabolism of these plant compounds, and hence the tea quality, is dependent on a variety of factors, including nitrogen supply (*Yang et al., 2018*).

Nitrogen (N), a crucial component for plant development, is frequently supplied *via* fertilizers to guarantee optimal growth. However, excess N inhibits the formation of flavonol glycosides, whereas decreasing N availability reduces amino acid and caffeine concentrations in mature tea leaves (*Li et al., 2016*; *Dong et al., 2019*). In addition, long-term nitrogen fertilization is not only expensive, but it also causes an array of environmental issues, including greenhouse gas emissions, soil pH changes, eutrophication, and microbial community disruption (*Gao & Cabrera Serrenho, 2023*; *Kamran et al., 2023*; *Liu et al., 2023*; *Tang et al., 2023*). Use of tea cultivars with high NUE (nitrogen uptake efficiency) and high quality is necessary to preserve environmental pollution and promote productivity.

Nitrogen-efficient varieties are likely to have polymorphisms in the genes that control nitrogen metabolism and determine the tea quality (*Li et al., 2017a*; *Li et al., 2017b*; *Li et al., 2017c*; *Yang et al., 2020*; *Xie et al., 2023*). Genes involved in N uptake (aquaporin PIP-type-like *PIP*, lysine histidine transporter 1-like *LHT1*), transport (ammonium transporter 1 member 2-like *AMT1.2*, high affinity nitrate transporter 2.4-like *NRT2.4*) and assimilation (alanine aminotransferase 2-like *AlaAT1*, glutamate dehydrogenase A *GDHa*, glutamate dehydrogenase 2 *GDH2*, glutamine synthetase nodule isozyme-like *GS1.2*) were identified as well as genes that regulate secondary metabolites which expression changes depending on the nitrogen level (*Wang et al., 2021b*; *Li et al., 2021*; *Xie et al., 2023*; *Wang et al., 2021a*; *Tang et al., 2021*; *Chen et al., 2023*; *Zhang et al., 2023*). Genes transcription factor MYB7-like and MYB4-like *(MYB7, MYB4)*, tryptophan-aspartic acid repeat protein repeat-containing protein HOS15-like (*WD40*), transcription factor bHLH35-like (*HLH35*), transcription factor bHLH36-like (*HLH36*), UDP-glycosyltransferase 71K2-like (*UFGTa*), anthocyanidin 3-O-glucosyltransferase 2-like (*UFGTb*), dihydroflavonol-4-reductase (*DFRa*), flavonoid 3′,5′-hydroxylase (*F3′5′Hb*) and flavonoid 3′,5′-hydroxylase 2-like

($F3'5'Ha$) are involved in the flavonoid pathway, whereas serine decarboxylase-like (*AlaDC*) controls the theanine synthesis (*Huang et al., 2018*; *Liu et al., 2018*; *Dong et al., 2019*; *Guo et al., 2019*; *Wang et al., 2021b*; *Ye et al., 2021*; *Li et al., 2023*). Gene Camellia sinensis 4-coumarate–CoA ligase-like 9 (*4Cl*) mediates phenylpropanoid metabolism, beta-glucosidase BoGH3B-like (*bG*) is critical for tea aroma generation, anthocyanidin reductase ((2S)-flavan-3-ol-forming)-like (*ANRb-ANR1*), leucoanthocyanidin dioxygenase-like (*ANSa* and *ANSb*), and leucoanthocyanidin reductase-like (*LAR*) regulate the catechin pathway, and WRKY transcription factor 57 (*WRKY57*) modulates stress responses (*Chen et al., 2009*; *Liu et al., 2015*; *Wani et al., 2021*; *Li et al., 2022*; *Zhao et al., 2022a*). In a previous work, we described 20 tea genotypes from Northwest Caucasia that are susceptible or tolerant to nitrogen deficit. A number of polymorphisms in the tea quality genes and their relationships with certain phenotypic traits as biochemical measurements were revealed in the tea collection (*Samarina et al., 2023*).

SNPs markers and numerous metabolic profile approaches could be utilized for identifying nitrogen-efficient cultivars (*Hazra et al., 2018*). The remote sensing technology provides a non-destructive and rapid approach to gauge plant health and development, offering insights into the metabolism change of tea plant response to nitrogen deficiency (*Cao et al., 2022*). The changes in plant chemical composition could be described by reflectance light-based indices or vegetation indices (VIs) developed based on the reflectance data (*Kior, Sukhov & Sukhova, 2021*). Combinations of spectral bands could be utilized for generating vegetation indices because pigments have the ability to absorb light in certain bands. Vegetation indices such as the water band index (WBI), photosynthetic rate index (PRI), normalized difference vegetation index (NDVI), transformed chlorophyll absorption in reflectance index (TCARI), triangular vegetation index (TVI), Zarco-Tejada & Miller Index (ZMI), flavanol reflectance index (FRI), and Anthocyanin Reflectance Index (ARI1, ARI2) provide information regarding plant water status, photosynthetic factors, and secondary metabolism, respectively (*Frels et al., 2018*; *Prey, Hu & Schmidhalter, 2020*). The use of vegetation indices to determine insect, cold, drought and nitrogen shortage stress enable the selection of the best growing conditions for tea plants (*Chen et al., 2021a*; *Chen et al., 2021b*; *Chen et al., 2021c*; *Zhao et al., 2022b*; *Mao et al., 2023*). Few research using unidentified aerial vehicles (UAVs) were conducted on the quality of tea and nitrogen deficiency (*Luo et al., 2022*). However, handled spectrometry was not tested to reveal the most efficient VIs for tea phenotyping, while experiments with potted plants rather than field studies are relevant for QTL and association mapping (*Hazra et al., 2018*).

In this study, we evaluate the efficiency of 31 VIs collected by a handheld spectrometer to reveal their efficiency for distinguishing ND-tolerant and ND-susceptible tea genotypes. We analyzed SNPs in 30 key genes related to N-assimilation and quality in the collection of 34 genotypes of tea plants in the Western Caucasus. We aimed to identify relationships between genotype and phenotype traits in ND-tolerant and ND-susceptible tea cultivars. The findings of the study could be used as markers for screening ND-tolerant tea genotypes. This research may advance precise breeding strategies aimed to enhance yield quality of *Camellia sinensis* (L.) O. Kuntze under ND by defining the genetic determinants and chemical composition linked to ND-response.

## MATERIALS & METHODS

### Plant material

The plant materials were obtained from the field gene bank of the Russian Academy of Sciences' Federal Research Center's Subtropical Scientific Center (FRC SSC RAS) (*Samarina et al., 2022*). This study comprised mutant forms obtained between 1970 and 1980 from seeds (mostly cultivars "Kolkhida" and "Qimen") exposed to $\gamma$-irradiation. Each genotype of plants was clonally reproduced using 30–60 replicates, and they were cultivated on acid soil from a brown forest (pH 5.5) with 30 mg kg$^{-1}$ of nitrogen (as opposed to the ideal 80 mg kg$^{-1}$ N for tea plantations). For the past 27 years, no fertilizers have been added to the experimental plot.

### Library preparation and amplicon sequencing

The library preparation and sequencing procedure for the following 14 genotypes of tea plants is explained; our earlier research on gene selection and primer design, long-range polymerase chain reaction, and sequencing for the remaining 20 variations can be reviewed in *Samarina et al. (2023)*.

Using the NEBNext Ultra II DNA Reagent Kit Library Prep Kit for Illumina and following the manufacturer's instructions, fragment DNA libraries were created equimolarly from the mixed PCR results. The libraries were subjected to a qualitative assessment with High Sensitivity D5000 ScreenTape and High kits Sensitivity D5000 Reagents (Agilent, Santa Clara, CA, USA) on an Agilent bioanalyzer TapeStation 4150. Using the KAPA Library Quantification Kit (KAPA Biosystems, Wilmington, MA, USA), a real-time PCR was used to provide a quantitative assessment of the products.

The DNA library fragments were mixed equimolarly into a pool and sequenced on the Illumina MiSeq using pair-end reads 76+76 bp and single-end reads 151 bp. Using the default settings of the bcl2fastq v2.20.0.422 software, sequencing data were demultiplexed by index sequences. For each DNA library, a total of 184,000–392,000 pairs of reads were collected. The FastQC v0.11.2 program was used to carry out the first quality evaluation of the deep sequencing data. Low-quality sequences and adapters were eliminated using AdapterRemoval v2 programs (with settings trimqualities, minquality 20, minlength 50). Following filtering, 94.34% of the read pairs were retained.

Data that had been filtered were mapped against the tea plant's reference genome (GCF_004153795.1). The BWA programs package's bwa mem function was utilized for mapping. Duplicates were eliminated using the MarkDuplicates function of the Picard tools v2.22.2 (Picard toolkit) software package. Samtools v1.9, a software application, was used to assess the alignments' quality. Using the COVERAGE_CAP = 10,000 option, the CollectWgsMetrics function of the Picard-tools software package (https://broadinstitute.github.io/picard/, accessed on March 2, 2024) was used to measure the depth coverage of the target genomic regions. 96.44% of reads on average were mapped to the genome of tea. On average, we were able to get 261-fold coverage of the target genomic areas of tea for each sample.

The raw data are deposited in the NCBI SRA database under accession numbers PRJNA1015448 and PRJNA977584.

## Genotype analysis

Using BWA-MEM (version 0.7.12), the clean reads were aligned to the reference genome "Shuchazao" (*Xia et al., 2020*), and SAMtools (version 1.16.1) was used for sorting and combination of paired-end and single-end reads into a single-bam file. The GATK software (version 4.2) was used to add read groups. Variant calling was done using the GATK-HaplotypeCaller method, with default parameters for diploid/unknown ploidy varieties and sample-ploidy 3 and sample-ploidy 4 for tetraploids and known triploids, respectively. The following parameters were utilized by the GATK software to select and filter SNPs/InDels: 'QD <2.0||FS >60.0||MQ <40.0||SOR_filter||SOR >4.0||DP <261' and 'QD <2.0||FS >200.0||SOR >10.0||DP <261', respectively.

SnpEFF (version 5.0) was used to build the database for the reference genome "Shuchazao", and it then served to annotate the remaining variants. High, moderate, low, or modifier effect impact classifications were obtained *via* the SnpEff tool variation annotation. These genetic differences known as impact variations are expected to have an indirect, mild, moderate, or severe effect on the protein.

In order to facilitate further study, the discovered SNP data of the 14 tea varieties were combined with published data on 20 tea sorts (Data S1). The formula for SNP density was mean SNP per gene divided by the gene's fragment length in kb. We normalized the SNP frequency in each gene to get a summary of the SNP distribution and potential SNP enrichments for the genes. Each SNP gene frequency was determined using the following formula:

$$\text{SNP\_freq} = (\text{SNP\_count/per\_gene})/\text{gene length} \times 10^3,$$ where gene_length is the length of the gene and SNP_count/per_gene is the number of SNPs found in a particular gene. To make a fair comparison more straightforward, the SNP_Freq values were leveraged by applying factor $10^3$ to the denominator.

## Phenotypic analysis

In this work the efficiency of 31 different VIs was evaluated to phenotype ND-response in tea collection. Using a Ci-710s Miniature Leaf Spectrometer (CID Bio-Science, Camas, WA, USA), the leaf spectral reflectance in the 350–1100 nm region was measured and 31 VIs were calculated. Five technical replications of each 33 genotypes were used to measure the reflectance in the middle of each leaf, next to the primary vehicle between 11:00 and 14:00. Data were statistically analyzed using the XLSTAT program (free trial version). To identify significant changes between the genotypes, one-way ANOVA, Fisher's and Tukey tests were performed. In addition, the study employed Pearson (n) PCA. The measured values of each VI as well as the results of statistical testing can be found in Data S2.

## Genotype and phenotype association analysis

For the association analysis, we combined SNP data from 20 and 14 different tea varieties. Locations of SNPs with moderate and high effect were mapped based on the alternative homozygous and heterozygous states of each allele (Data S3). Only 26 types were subjected to a further study since phenotype data were available for a portion of the genotypes that had been sequenced. To determine the relationships between SNPs and the phenotypes, a linear

regression model was combined with a statistical test adjusted for multiple comparisons (Bonferroni and false discovery rate (FDR)). Significant associations were identified at Bonferroni- and FDR-corrected $p$-values <0.05. Statistical analysis and visualization were performed using R package (version 4.2.3; *R Core Team, 2023*).

## RESULTS

### Phenotypic characterization

To reveal efficient VIs for phenotyping of ND- response, tea genotypes were classified as tolerant or susceptible to ND based on their leaf quality and leaf N-content. Eight genotypes were assigned as tolerant of ND, whereas the other ten were assigned as susceptible. Other fourteen genotypes did not exhibit any clear response to ND and were classified as non-responsive.

To illustrate the correspondence of genotypes, phenotypic traits and vegetation indices PCA biplot was used (Fig. 1). The first two PCs displayed a cumulative variation of approximately 73.03%. Both the ND-susceptible and ND-tolerant genotypes were clearly separated in the biplot. Most of the vectors of VIs were distributed with high loading on the positive side of PC1 and the negative side of PC2. The highest loading was observed in the following VIs: ZMI, VREI1, RENDVI, CNDVI, PRI, PSRI, GM2, GM1, and CRI2, NDVI, SIPI, and CRI1 indicating their positive correlation. The majority of the ND-tolerant genotypes were distributed close to these VI, suggesting positive correlation between these indices and ND-tolerance. In contrast, genotypes with no clear response to ND were placed on the negative side of PC2. The vectors of TCARI, Ctr2, VREI2, VREI3, and MDATT were positioned on the negative side of PC1, while Lic2, SRPI, and MRESRI —on the positive side of PC2. The majority of ND-susceptible genotypes were placed closely to them, having greater values of these VIs as compared to ND-tolerant ones. Finally, few ND-tolerant accessions were placed in different PCA-sides.

Those VIs which showed coefficients of determination $R^2 > 0.5$ and $p$ values < 0.0001 were assigned as efficient for selection of ND-tolerant tea accessions (Table 1). Based on Tukey's multiple comparisons the following VIs showed greater values into tolerant genotypes as compared to susceptible ones: ZMI, CNDVI, RENDVI, VREI1, GM2, GM1, PRI, PSRI, PRI, ARI2, ARI1, WBI, NDVI, SIPI, Lic1, and WBI. On the contrary, ND-susceptible genotypes displayed higher values for MDATT, Ctr2, TCARI, MCARI1, VREI3, and VREI2 as compared to tolerant ones. According to the prediction analysis, the greatest distance between susceptible and tolerant groups was observed by ZMI, RENDI and CNDVI (Fig. 2). Tolerant genotypes showed ZMI values above 1.9, while susceptible-1.7. RENDVI and CNDVI were below 1.35 for susceptible, and above 1.40 for tolerant genotypes. Additionally, the remarkable differences were observed by PRI, GM1, GM2 and VREI1, and tolerant genotypes displayed greater values. In contrast, ND-susceptible genotypes showed larger values of VREI3, VREI2, and Ctr2, which were above −0.07, −0.07 and 0.25, respectively.
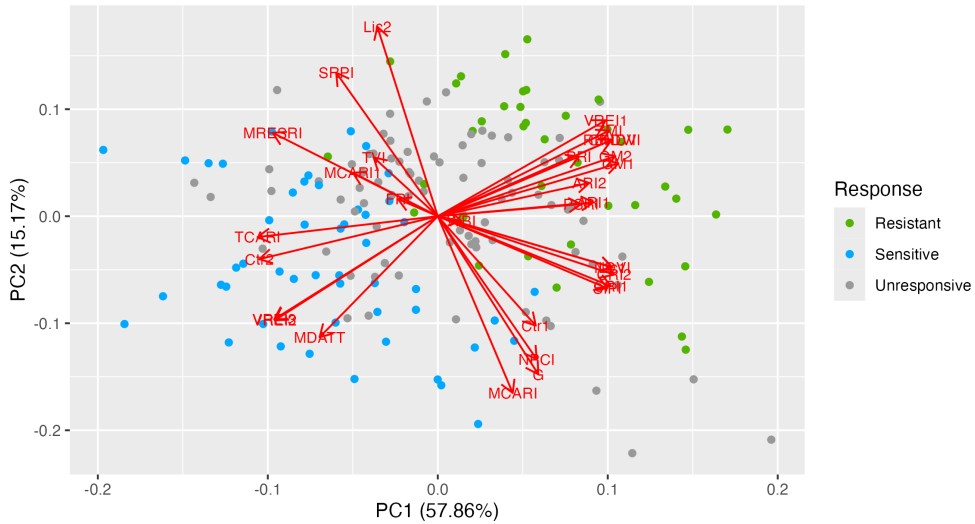

**Figure 1** **Principal component analysis of 31 vegetation indices in 33 tea genotypes with different responses to nitrogen deficit.** PCA biplot illustrates the related traits and correlations between vegetation indices and genotypes. The first two PCs exhibited a cumulative variation of approximately 73.03%. Both the ND-susceptible and ND-tolerant genotypes were clearly separated in the biplot. Most of the vectors of VIs were distributed with high loading on the positive side of PC1 and the negative side of PC2. The highest loading was observed in the following VIs: ZMI, VRE1, RENDVI, CNDVI, PRI, PSRI, GM2, GM1, and CRI2, NDVI, SIPI, and CRI1.

## Identification of SNPs in tea quality genes and their associations with phenotypes

Among 34 tea accessions, *4CL, AMT1,2,* and *F3′5′Hb* showed the highest SNPs densities (1.0–2.0) in exon regions, while *AlaAT1, GDH2, LAR, WD40, bG,* and *bHLH35* showed the lowest densities (Table 2). The highest SNPs-densities in introns (5.0–6.0) was found in *4CL* and *GS1,2*. There were no SNPs found in *MYB7* or *bHLH36*. The largest percentage of polymorphisms in exon per gene (more than 45%) were detected for *bG, F3′5′Hb,* and *DFRa*.

The high-effect-SNPs were observed in the following accessions: #619, #2697, #536, #1385 and #3986 (Fig. 3). Low-effect SNPs were found to have the highest percentages in #582, #157, and cv. Karatum, ranging from 4.0 to 25.0% across all genotypes. In cv. Sochi, #35, and #1292, moderate-effect SNPs have the highest rate, varying from 5.0 to 15.9% across all genotypes. The highest percentages of SNPs with modifying effects were detected in #321, #619, and #3509, and ranged from 63.0 to 86.7% across all genotypes.

The intron variants were the most frequent SNPs (8.9–57.14%) across all genotypes with the highest rate in ND-susceptible genotypes #551, #507, and #1467 –(Fig. 3, Data S4). The highest percentage of intragenic variations SNPs (56–67%) was observed in #321, #35, while the lowest (1.5–4.5%) in showed #619, #1385 The highest values of intergenic region SNPs (9–11%) were detected in #321 and #3823, while the lowest (0.5–0.8%) in cv. Sochi and #837. Generally, the lowest SNP-frequencies were observed for 3′-UTR (0.6–8.0%), 5′-UTR (0.5–6.38%), and splice region or accepter variations (0.2–2.13%). The highest

**Table 1  Determination coefficients of vegetation indices at *p* value <0.05 and determination coefficient R2 > 0.5.**

| VI | R2 | F | Pr >F |
|---|---|---|---|
| CNDVI | 0.728 | 11.487 | <0.0001 |
| RENDVI | 0.728 | 11.487 | <0.0001 |
| SIPI | 0.479 | 3.928 | <0.0001 |
| NDVI | 0.528 | 4.792 | <0.0001 |
| MDATT | 0.624 | 7.102 | <0.0001 |
| Lic1 | 0.528 | 4.792 | <0.0001 |
| Ctr2 | 0.683 | 9.224 | <0.0001 |
| ARI2 | 0.652 | 8.022 | <0.0001 |
| TCARI | 0.635 | 7.451 | <0.0001 |
| MCARI1 | 0.548 | 5.198 | <0.0001 |
| WBI | 0.624 | 7.115 | <0.0001 |
| ZMI | 0.744 | 12.437 | <0.0001 |
| VREI1 | 0.759 | 13.501 | <0.0001 |
| GM2 | 0.695 | 9.757 | <0.0001 |
| GM1 | 0.674 | 8.849 | <0.0001 |
| PSRI | 0.603 | 6.508 | <0.0001 |
| PRI | 0.674 | 8.864 | <0.0001 |
| VREI3 | 0.770 | 14.357 | <0.0001 |
| VREI2 | 0.771 | 14.333 | <0.0001 |
| ARI2 | 0.625 | 7.136 | <0.0001 |

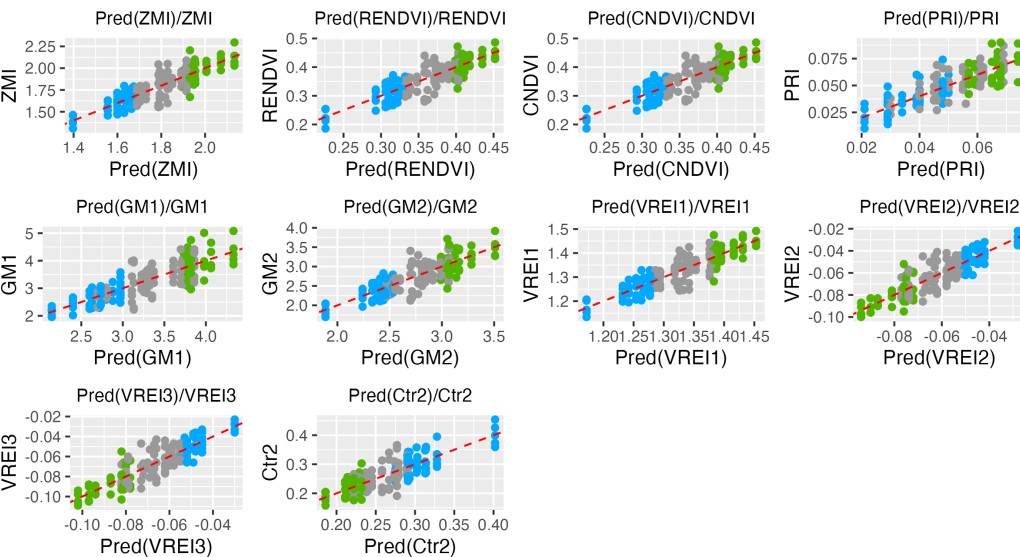

**Figure 2  Data points distributions and prediction in the tea genotypes with different responses to nitrogen deficit.** Green, tolerant genotypes; blue, susceptible genotypes; gray, non-responsive.

**Table 2 The distribution of SNPs in 34 distinct varieties of tea in the exon and intron regions of the target genes ($N = 20$).**

| Gene | Fragment Length, bp | Mean SNPs Number in Introns | Mean SNPs Number in Exons | SNP Density in Introns | SNP Density in Exons | SNP % in Exons |
|---|---|---|---|---|---|---|
| *4CL* | 5264 | 32.94 | 9.17 | 6.26 | 1.74 | 21.78 |
| *AMT1,2* | 2643 | 5.40 | 4.29 | 2.04 | 1.62 | 44.25 |
| *AlaAT1* | 8058 | 1.06 | 0.03 | 0.13 | 0.00 | 2.63 |
| *AlaDC* | 7227 | 6.83 | 2.97 | 0.94 | 0.41 | 30.32 |
| *DFRa* | 6600 | 2.54 | 2.49 | 0.39 | 0.38 | 49.43 |
| *F3′5′H_a* | 5118 | 6.57 | 1.86 | 1.28 | 0.36 | 22.03 |
| *F3′5′Hb* | 4435 | 3.66 | 4.54 | 0.82 | 1.02 | 55.40 |
| *GDH2* | 4915 | 2.57 | 0.11 | 0.52 | 0.02 | 4.26 |
| *GS1,2* | 6202 | 33.11 | 2.77 | 5.34 | 0.45 | 7.72 |
| *LAR* | 8600 | 0.31 | 0.06 | 0.04 | 0.01 | 15.38 |
| *LHT1* | 5107 | 2.00 | 0.80 | 0.39 | 0.16 | 28.57 |
| *MYB4* | 5342 | 5.00 | 0.60 | 0.94 | 0.11 | 10.71 |
| *MYB7* | 3376 | 0.03 | 0.00 | 0.01 | 0.00 | 0.00 |
| *NRT2,4* | 3060 | 6.03 | 2.37 | 1.97 | 0.77 | 28.23 |
| *PIP* | 2006 | 1.37 | 0.83 | 0.68 | 0.41 | 37.66 |
| *WD40* | 3844 | 2.34 | 0.34 | 0.61 | 0.09 | 12.77 |
| *WRKY57* | 11214 | 23.91 | 2.46 | 2.13 | 0.22 | 9.32 |
| *bG* | 8605 | 0.11 | 0.37 | 0.01 | 0.04 | 76.47 |
| *bHLH35* | 5743 | 1.91 | 0.03 | 0.33 | 0.00 | 1.47 |
| *bHLH36* | 2953 | 24.23 | 0.00 | 8.20 | 0.00 | 0.00 |

rates of 5′-UTR SNPs were observed in #582, cv. Karatum, and #551, while the lowest were observed in #Sochi and #4605. On the other hand, the highest percentage of 3′-UTR SNPs was detected in #1292, #1385, while the lowest was detected in cv. Karatum, #1476, #837. Splice areas and splice acceptor variations were rare observed in #3986 and #619 and are predominantly occurring in cv. Karatum, #582, and #3823. The downstream and upstream gene variations were ranged as 5.6–21.38% and 0.5–18%, respectively. The greatest values were detected in #619, #3180, #855,#257, and #501, while the lowest were in #551, #3823, and #527.

The highest exon SNPs frequency was observed in *UFGTa*, *4Cl*, *UFGTb,* and *AMT1.2*, while the lowest was in *GDH2, WD40, bHLH35, AlaAT1, LAR, GDHa*. The hierarchical clustering indicated no clear separation of genotypes by ND-tolerance, each branch combined both tolerant and susceptible accessions (Fig. 4). The first branch consisted of four tea genotypes with the highest SNP-frequencies in *UFGTa*: #507, #1476, #1484 and #Sochi. Among them, ND-susceptible #507 displayed the lowest leaf N-content, #1476 was ND-tolerant with high leaf N-content, and #1484 and #Sochi showed an uncertain reaction to ND. The second branch consisted of the two sub-branches. The first sub-branch combined the ND-susceptible genotypes with the low leaf nitrogen content, namely #1385, #3986, #1467, #582, #527, #1877, and #536. In addition, ND- tolerant genotype #619

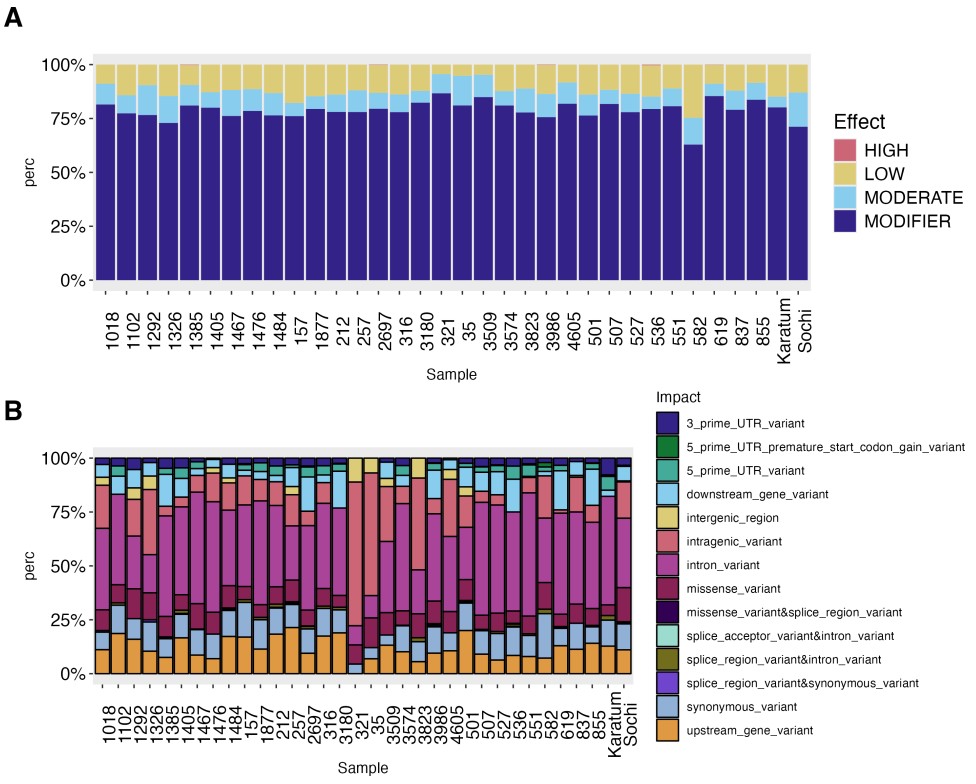

**Figure 3** **SNPs effect on the phenotypes of 34 tea genotypes.** (A) SNPs effect on the phenotypes of 34 tea genotypes. (B) SNPs impact on the phenotypes of 34 tea genotype.

and the high nitrogen-content genotypes #316, #212, and #1405 were joined to this sub-branch. All these tea plant genotypes displayed significant SNP frequencies in *NRT2.4*, *PIP*, *AlaDC*, *DFRa*, *GS1.2*, *F3′5′Hb*, *UFGTa*, *UFGTb*, *4Cl*, and *AMT1.2*. The second sub-branch combined ND-susceptible genotypes (#501, #551), ND-tolerant genotypes (#157, #2697, #3609, #4605) and non-responsive to ND.

Totally, 109 SNPs were classified as missense variations causing amino acid changes with a moderate effect (Data S5). A single SNP in *WRKY57* with a significant effect was identified as a splice acceptor and intron variant in ND-susceptible genotypes #3986 and #1385, as well as ND-tolerant genotypes #619, #2697, and #536. The most frequent amino acid alterations were revealled in *4CL*, *F3′5′Hb*, *F3′5′Ha* and *ANRb-ANR1*. A number of SNPs specific for ND-susceptible genotypes and genotypes with low N content (#855, #3574, and #536) was revealed. These mutations lead to amino acid changes in *AlaAT1*, *MYB4*, and *WRKY57*.

Finally, four significant associations (*p* value < 0.05) were revealed between the SNPs and vegetation indices (Table 3). Two SNPs in *4Cl* were associated with the water band index (WBI), with a significant coefficient of determination ($R^2 = 0.624$). Both SNPs of the *4Cl* were occurred in #1292 and the ND-susceptible genotype #507. Additionally, associations between FRI ($R^2 = 0.211$) and SNPs that alter the amino acid composition of *PIP* and *ANSb* were found. The SNP in *ANSb* was observed in #619, #157 (ND-tolerant

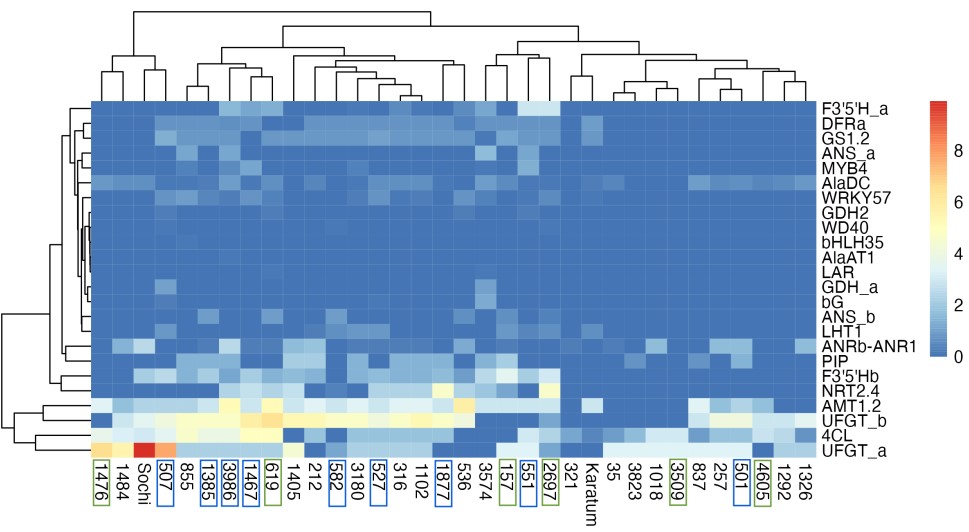

**Figure 4** **Heatmap of exon SNPs frequencies.** The columns represent the tea genotypes, and the rows represent the different genes. Green frames indicate ND-tolerant genotypes, blue frames indicate ND-susceptible genotypes.

**Table 3** **Significant associations between SNPs and the phenotypes at *p* value <0.05.**

| Gene | Position | N (REF/ALT) | VI | Amino (REF/ALT) | Property (REF/ALT) | Test Statistics | DF | Exact *P*-value | Adj. *P*-value |
|------|----------|-------------|-----|-----------------|--------------------|-----------------| ---|-----------------|----------------|
| *4CL* | 2130421 | A/T | WBI | p.Thr16Ser | Neutral/Neutral | 4.188505 | 23 | 0.0003519712 | 0.03554909 |
| *4CL* | 2132938 | A/G | WBI | p.Ile417Val | Hydrophobic/Hydrophobic | 4.188505 | 23 | 0.0003519712 | 0.03554909 |
| *PIP* | 220018 | A/T | FRI | p.Val182Glu | Hydrophobic/Charged_acidic | −7.629861 | 14 | 2.366524e−06 | 0.0002390189 |
| *ANSb* | 805981 | A/G | FRI | p.Phe39Leu | Aromatic/Hydrophobic | −7.32134 | 13 | 5.820357e−06 | 0.000587856 |

**Notes.**
N (REF/ALT), nucleotide change (reference/alternative); VI, vegetation indice; DF, Degrees of Freedom; Adj. *P*-value, Bonferroni-corrected *P*-value.

genotypes), #582, #1385 and #536 (ND-susceptible genotypes). The mutation in the *PIP* gene was found in #157 (ND-susceptible) and #212 (ND-tolerant) genotypes.

## DISCUSSION

This study was aimed to search functional SNPs and efficient vegetation indices in the collection of tea plant *Camellia sinensis* (L.) Kuntze. We used the field tea gene bank of Western Caucasus grown without nitrogen fertilizers. Our earlier study reported the significant level of genetic diversity in the studied tea collection (*Samarina et al., 2022*). Controlled hybridization, $\gamma$-irradiation, and clonal selection were used to create this tea gene bank characterized by number of valuable horticultural traits.

This study was the first to employ portable spectrometry to reveal efficient vegetation indices (VIs) for phenotyping of ND-tolerant tea plants. A total of 20 of 31 VIs showed to be efficient for ND-response phenotyping. Also, prediction analysis indicated the greatest gap for ZMI, RENDI, CNDVI, PRI, GM1, GM2, VREI1 (tolerant genotypes have higher values) and VREI3, VREI2, Ctr2 (susceptible genotypes have higher values),

suggesting that these are the most reliable VIs for ND-response phenotyping. These VIs are sensitive to chlorophyll concentration and nitrogen stress (*Penuelas, Baret & Filella, 1995*; *Lichtenthaler et al., 1996*; *Haboudane et al., 2004*; *Jain et al., 2007*; *Sun et al., 2013*; *Burns et al., 2022*; *Vogelmann, Rock & Moss, 1993*). One of the main traits of tea plant adaptability is the amount of chlorophyll in the leaves, which rises directly with the amount of nitrogen applied (*Qiu et al., 2024*). Chlorophyll preservation strategies could be efficient strategy to developND- tolerant genotypes to. Nitrate levels correspond to VIs that are sensitive to chlorophyll concentration, such as Ctr2, NDVI, RENDVI, and TCARI (*Katsoulas et al., 2016*; *Ihuoma & Madramootoo, 2020*). PRI, which was higher in ND-tolerant tea genotypes, describes the intensity of photosynthesis based on the amount of chlorophyll (*Xiao et al., 2018*). Additionally, the carotenoid pigment-sensitive indicator PSRI and the anthocyanin reflectance indices ARI1/ARI2 indicate plant senescence or active growth and were efficient to select ND-tolerant tea genotypes (*Merzlyak et al., 1999*; *Gitelson, Merzlyak & Chivkunova, 2001*; *Foster et al., 2012*; *Tayade et al., 2022*).

This corresponds with the suggestion that genotypes withelevated N content also characterized byelevated levels of polyphenols, specifically flavonols, which can be detected using PSRI and ARI1/ARI2. Long-term N fertilization increases carotenoid concentration in tea leaves, while ND promotes oxidative stress in plants (*Chen et al., 2021b*). Furthermore, it was shown that anthocyanins and carotenoids are accumulated under weak stresses and promote the antioxidant process (*Stahl & Sies, 2003*; *Xiang et al., 2022*). Thus, it can be suggested that oxidative stress-protective mechanisms are triggered in ND-tolerant tea genotypes (*Peñuelas et al., 1994*; *Badzmierowski, McCall & Evanylo, 2019*). Some researchers demonstrated a significant increase in water use efficiency with increasing leaf N content (*Katsoulas et al., 2016*). This corresponds with our findings on correlation of photosynthetic efficiency, biomass, nitrogen and water content related indices in ND-tolerant genotypes. Thus, these VIs can be used for selection of ND-tolerant tea genotypes. Other VIs showed no difference between ND-susceptible and ND-tolerant genotypes.

Association analysis revealed four SNPs causing amino changes in the N-metabolism related genes. The WBI was associated with two SNPs in *4Cl*, encoding 4-coumarate:CoA ligase and involved in the phenylpropanoid biosynthesis pathway (*Li et al., 2022*). In addition, SNP in *4Cl* was associated with the antioxidant polyphenol theaflavin. Flavonoids and polyphenols are known for their role in defense against biotic and abiotic stressors including water stress. Water stress has been shown to be a cause of phenolic compound formation, and a decrease in soil water content lowers the phenols content in tea (*Cheruiyot et al., 2007*; *Hodaei et al., 2018*). Consequently, WBI has the potential to be used as an indirect indicator of phenylpropanoid leaf content. Changes of water and polyphenol contents in leaves could be better understood by investigating how ND-efficient tea genotypes react to water stress. In addition, two SNPs change amino acids with similar properties (Thr to Ser and Ile to Val), which could have a minor impact on the enzyme structure and functions. SNPs in the *ANSb* and *PIP* showed positive association with FRI (flavonol reflectance index) (*Merzlyak et al., 2005*). Anthocyanins are phenolic compounds synthesized and accumulated by anthocyanidin synthase, which is encoded by *ANSb*

(*Anggraini et al., 2019*; *Huang et al., 2022*). Moreover, the anthocyanin content is affected by the increased production of ROS by plasma membrane intrinsic proteins (PIPs), which also participates in N uptake (*Li et al., 2017a*; *Zhang et al., 2020a*; *Maritim et al., 2021a*). Despite the fact, that FRI showed lowR2, yet the association between SNP and FRI is evident. In our study, the phenotypic data was available for a portion of the genotypes only, whereas the SNPs-data was obtained from the two combined studies. This could have caused some gaps in the data and affected the findings of the association study.

According to the SNPs- analysis, the genes controlling ammonium transport (*AMT1.2*) and flavonoid pathways (*UFGTa, UFGTb, 4Cl, F3′5′Hb, ANRb-ANR1*) showed the highest SNP frequencies across all genotypes. Other studies reported that, SNPs in *4Cl*, *F3′5′H*, *DFR*, *LAR*, *ANS*, and *ANR* in cultivars 'Shuchazao' and 'Yunkang 10' affected catechin/caffeine contents (*Liu et al., 2019*; *Zhang et al., 2020b*), however we revealed no relationship between the amount of N and the total catechin content. This is in accordance to the study, which showed no relationship between the amount of N and the quantity of catechins, synthesized by *ANR* and *4Cl* in tea leaves (*Zhang et al., 2020b*). However, a strong positive association was found between the leaf N-content and flavanols content, specifically theaflavins, and thearubigins, as well as tannins like gallic acid. In recent study, SNPs related to the synthesis of phenylpropanoid/flavonoid were found in the *ANR1*, *LAR*, *F3′5′Hb*, *4Cl*, *UFGTa*, and *UFGTb* genes across multiple genotypes; however, their relationship with ND was not studied (*Maritim et al., 2021b*). *Jiang et al. (2020)* showed that SNP within the chalcone synthase (CHS) gene was shown to be functionally associated with catechin content. Recently we revealed that one SNP in *4Cl* was significantly associated with Theaflavin content (*Samarina et al., 2023*). *Fang et al. (2021)* revealed 17 SNPs that were significantly or extremely significantly associated with specific metabolite levels.

Earlier research has also demonstrated that the accumulation of flavonoids by ND-tea plants positively correlated with increased expression of *F3H*, *FNS*, *UFGT*, *bHLH35*, and *bHLH36* (*Huang et al., 2018*). Additionally, increased expression of dihydroflavonol 4-reductase (*DFR*), anthocyanidin synthase (*ANS*), anthocyanidin reductase 1 (*ANR1*), and 3′,5′-hydroxylase (*F3′5′H*) were revealed under N excess as compared to ND (*Dong et al., 2019*). Further research using high-performance liquid chromatography is required to demonstrate the leaf content of proanthocyanidins and how it relates to N content. A group of ND-susceptible genotypes and several ND-tolerant genotypes have more SNPs in *NRT2.4*, *PIP*, *AlaDC*, *DFRa*, and *GS1.2*. Nitrogen accumulation and *NRT2.4* SNPs were positively correlated in the study of tea germplasms from Shandong Province (*Fan et al., 2022*). The aquaporin gene (*PIP*) is also responsible for effective N uptake, while Alanine decarboxylase (*AlaDC*) is crucial for nitrogen storage participating in theanine synthesis (*Wang et al., 2021b*; *Xie et al., 2023*; *Bai et al., 2019*; *Bai et al., 2021*). L-theanine pathway and ammonium assimilation are facilitated by glutamine synthetase (*GS1.2*) (*Zhang et al., 2023*). For instance, SNPs connected to theanine biosynthesis were discovered in GS1.2 in the Indian tea collection (*Maritim et al., 2021b*). Recently, the preliminary association analysis showed that two SNPs (CsSNP07 and CsSNP11) within CsNRT2.4 were significantly associated with nitrogen accumulation (*Fan et al., 2022*). In their study,

35 tea genotypes were analyzed and 46 SNPs were revealed within genes involved in nitrogen uptake, assimilation, and allocation.

Recently, *Guo et al. (2023)* revealed two alleles of *CsGS* (*CsGS*-L and *CsGS*-H) which overexpression enhanced the contents of glutamate and arginine in transgenic plants. They found $SNP_{1054}$ which is important for CsGS catalyzing glutamate into glutamine. Furthermore, *CsGS*-L and *CsGS*-H differentially regulated the accumulation of glutamine. In our study, SNPs in the abovementioned genes probably involved in significant variations in the chemical contents in leaves; #316 showed the highest theanine and nitrogen content, whereas #1467, #1877, #527, #536, and #507 showed the lowest. SNPs that change amino acids in the *AlaAT1* and *MYB4* were specific to ND-susceptible tea genotypes and those characterized by the low leaf N-content. Alanine aminotransferase *AlaAT* plays a role in the biosynthesis and accumulation of L-theanine as well as the efficiency of nitrogen use (*Wang et al., 2021a*; *Zhang et al., 2022*). Thus, we suggest that this alteration in the structure of the enzyme result in L-theanine decrease in *#3986* and *#1467*. The low leaf N-content, were positively correlated with flavan-3-ols and other phenolic compounds whose accumulation is inhibited by *MYB4* (*Li et al., 2017b*; *Ye et al., 2021*). Finally, a single SNP in *WRKY57* was identified in ND-susceptible genotypes. This transcription factor participates in ABA-mediated stress responses, (*Jiang et al., 2014*; *Chen et al., 2019*; *Chen et al., 2021c*). However, the role of *WRKY57* in nitrogen stress has yet to be investigated. Combining datasets under different experimental settings presents data integration challenges that could impair accuracy and result in missing values in SNPs positions (*Dergilev et al., 2021*; *Chao et al., 2023*). Further phenotype studies and Sanger sequencing has to be applied to validate the results. Another limitation of this study is the small sample size, which does not allow to calculate linkage disequilibrium (LD). Further characterization of tea varieties cultivated under ND-conditions, as well as the validation using sequencing and metabolic techniques, could improve the accuracy of detecting genotypes that are tolerant or susceptible to ND.

## CONCLUSIONS

We identified efficient vegetation indices to distinguish ND-tolerant and ND-susceptible tea genotypes: ZMI, RENDI, CNDVI, PRI, GM1, GM2, VRI1, VRE3, VRE2, Ctr2. Numerous SNPs that could be exploited for genotyping were discovered. Among them, mutations in *NRT2.4, PIP, AlaDC, DFRa, GS1.2, AlaAT1, MYB4*, and *WRKY57* were specific for ND-susceptible tea genotypes. Four associations between the SNPs and vegetation indices were identified. Particularly, water band index (WBI) and flavonol reflectance index (FRI) were associated with SNPs in the flavonoid regulators *4Cl*, *ANSb*, and *PIP*. The phenotypic and genetic data obtained in this study could be used in breeding programs aimed at developing nitrogen-efficient tea cultivars.

## ACKNOWLEDGEMENTS

All phenotypic analyses were conducted at Subtropical Scientific Center. All genetic analyses were conducted at Sirius University of Science and Technology. The plant material for this study was provided by the program FRC SSC RAS # FGRW-2024-0003. Data analysis was partly done at Sechenov University.

### Funding

This study was funded by Russian Science Foundation grant #22-16-00058. The funders had no role in study design, data collection and analysis, decision to publish, or preparation of the manuscript.

### Grant Disclosures

The following grant information was disclosed by the authors:
Russian Science Foundation: #22-16-00058.

### Competing Interests

Yuriy Orlov is an Academic Editor for PeerJ.

### Author Contributions

- Daria Kuzmina conceived and designed the experiments, performed the experiments, prepared figures and/or tables, authored or reviewed drafts of the article, and approved the final draft.
- Lyudmila S Malyukova performed the experiments, prepared figures and/or tables, and approved the final draft.
- Karina Manakhova performed the experiments, prepared figures and/or tables, and approved the final draft.
- Tatyana Kovalenko performed the experiments, prepared figures and/or tables, and approved the final draft.
- Jaroslava Fedorina performed the experiments, prepared figures and/or tables, and approved the final draft.
- Aleksandra O. Matskiv performed the experiments, prepared figures and/or tables, and approved the final draft.
- Alexey V. Ryndin analyzed the data, authored or reviewed drafts of the article, and approved the final draft.
- Maya V. Gvasaliya analyzed the data, authored or reviewed drafts of the article, and approved the final draft.
- Yuriy L. Orlov analyzed the data, authored or reviewed drafts of the article, and approved the final draft.
- Lidiia S. Samarina conceived and designed the experiments, analyzed the data, prepared figures and/or tables, authored or reviewed drafts of the article, and approved the final draft.
## Data Availability

Raw data, including tea plant leaves measurements, are available in the Supplemental Files.

## Supplemental Information

Supplemental information for this article can be found online at http://dx.doi.org/10.7717/peerj.17689#supplemental-information.

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
