# Peer review of "Associations between SNPs and vegetation indices: unraveling molecular insights for enhanced cultivation of tea plant (Camellia sinensis (L.) O. Kuntze)"

_PeerJ, doi:10.7717/peerj.17689_

## Round 0.1 · original submission · Minor Revisions

Dear Dr. Orlov,

Your article has been reviewed by three independent reviewers. Both the reviewers and myself, consider that the work is scientifically sound, providing considerable advances to plant science, in general, and tea research, in particular. Despite that, I suggest minor revisions to improve the quality os the manuscript. Below you can find the reviewers' comments, from which I highlight:

- Results: can be improved to increase clarity

- Discussion: pinpoint the significance of the results for te research and plant science; incorporate more literature on SNP analysis of tea plants and discuss data accordingly

- Figures: check the resolution

Looking forward to receiving the revised manuscript,

Sincerely
Ana I. Ribeiro-Barros

**Language Note:** PeerJ staff have identified that the English language needs to be improved. When you prepare your next revision, please either (i) have a colleague who is proficient in English and familiar with the subject matter review your manuscript, or (ii) contact a professional editing service to review your manuscript. PeerJ can provide language editing services - you can contact us at copyediting@peerj.com for pricing (be sure to provide your manuscript number and title). – PeerJ Staff

Reviewer 1 ·

Basic reporting

The study successfully identified efficient vegetation indices such as ZMI, RENDI, CNDVI, PRI, GM1, GM2, VRI1, VRE3, VRE2, and Ctr2. These indices help distinguish between N-deficiency(ND) tolerant and susceptible tea genotypes.
The research discovered numerous single-nucleotide polymorphisms (SNPs) that could be exploited for genotyping. This finding provides valuable genetic markers for further studies and breeding programs.
Gene mutations such as NRT2.4, PIP, AlaDC, DFRa, GS1.2, AlaAT1, MYB4, and WRKY57 were associated with ND-susceptible tea genotypes. This insight into the genetic factors underlying susceptibility to ND could aid in targeted breeding efforts.
The study detected four associations between SNPs and vegetation indices. Notably, associations were found between the water band index (WBI) and the far red index (FRI) with SNPs in flavonoid regulators 4Cl, ANSb, and PIP. This suggests a potential link between genetic variations and physiological traits related to nitrogen deficiency.

Experimental design

Phenotypic spectrometric analysis, next-generation sequencing, genotyping and QTL analysis are used.
The integrated approach provided high-quality sequencing data and facilitated the analysis of genomic variation and traits in tea plants.

Validity of the findings

The raw data was deposited in the NCBI SRA database under accession numbers PRJNA1015448
(https://www.ncbi.nlm.nih.gov/sra/SRX21783698) and PRJNA977584.

The results obtained are adequate and can be used for genotyping other specimens.

Additional comments

Overall, the research presents valuable insights into the genetic and physiological factors influencing tea genotype responses to nitrogen deficiency, with implications for research and practical breeding applications.

·

Basic reporting

The first abbreviation should be include full name for PIP, LHT1, etc.

Experimental design

The SNPs analyzed in this study are from the known genes, but the sequence is Whole Genome Sequencing. Thus, how to get the sequences for each known gene?

Validity of the findings

Because there is no validation of SNP loci associated with Vegetation Indices, at least, it should be present its genotype in the studied materials in the discussion.
Key figures on identification of SNP loci associated with vegetation indices, such as linkage disequilibrium, LD.

Reviewer 3 ·

Basic reporting

Overall, the English language usage in this manuscript is quite fluent, meeting the standards for academic publications. The article employs specialized scientific terminology and adheres to the structural and guideline norms of scientific research reporting. However, it should be noted that the results section could be presented more clearly and accurately. The discussion should elucidate the significance of these results and establish a stronger connection with the existing literature on SNP analysis of tea plants (Camellia sinensis).
The references throughout the article provide an adequate background on studies related to the genetics and phenotypic research of tea plants, nitrogen use efficiency, biosynthesis of secondary metabolites, and the impact of nitrogen deficiency on plant physiological and biochemical characteristics. The authors demonstrate a deep understanding of current research trends and existing knowledge. Additionally, by citing the latest research, the article underscores its relevance and timeliness.
The structure and figures of the article appear professional, with designs that are both professional-looking and informative, aiding in the visual presentation of the research findings. However, the resolution of the figures is relatively low, and attention should be paid to whether they meet the journal’s requirements.

Experimental design

The research objective of the article is to identify functional single nucleotide polymorphisms (SNPs) within key genes associated with nitrogen assimilation in tea plants (Camellia sinensis) and to determine the potential associations between these SNPs and phenotypic traits. The study employed spectral reflectance measurements and vegetation indices to assess the phenotypic response of tea plants, which are non-destructive methods suitable for large-scale field studies. SNPs were identified through pooled amplicon sequencing, annotated, and effect predictions were made using the SnpEFF tool, a common approach in molecular biology research. A linear regression model was applied to reveal the association between functional SNPs and efficient vegetation indices, a statistically sound method for exploring the relationship between genetic variation and phenotype. The experimental design of the article is reasonable and meets the requirements for clarity, relevance, significance, and filling knowledge gaps in the research question. These experiments hold promise for providing new insights and tools for molecular breeding and nitrogen management in tea plants. However, the study only used two cultivars, “Kolkhida” and “Qimen”, and the selection of tea plant varieties and growth conditions should represent the scope of the research question. If the experiment is limited to specific varieties or environments, it may not be applicable to a broader context.

Validity of the findings

The manuscript combines molecular markers and spectral technology to study the response of tea plants to nitrogen deficiency, representing a relatively novel research area that merges molecular biology with remote sensing techniques. The study has identified specific SNPs and vegetation indices related to nitrogen use efficiency, which may contribute to the development of new tea plant varieties, enhancing the sustainability and efficiency of tea production. The research could provide a supplement to existing knowledge, particularly regarding the molecular mechanisms of tea plant responses to nitrogen deficiency. However, it is necessary to evaluate to what extent these findings alter the current understanding and further clarify this in the discussion section.

---

## Round 0.2 · accepted · Accept

Dear Dr. Olov,

I am pleased to inform you that your manuscript is now accepted to be published in PeerJ. Nevertheless, please address the last editing details, highlighted in yellow in the attached version.


·

Basic reporting

The manuscript has been revised by my review's comments and suggestions.

Experimental design

It is good for its results.

Validity of the findings

The study got some interesting results.